# The NTR/prodrug revolution: Tools for controlling cell loss and regeneration

**Gha-Hyun J Kim[1,2], Michael Parsons[3]***

[1]Department of Clinical Pharmacy Practice, UC Irvine School of Pharmacy and Pharmaceutical Sciences, Irvine, United States; [2]Robert A. Mah Molecular Innovation Center, University of California, Irvine, Irvine, United States; [3]Department of Developmental and Cell Biology, School of Biological Sciences, Natural Sciences II, University of California, Irvine, Irvine, United States

## eLife Assessment

This Review Article nicely synthesizes the development, applications, and recent technical advances of the nitroreductase/prodrug system, highlighting how it enables precise spatio-temporal cell ablation and experimental platforms for studying regenerative mechanisms and screening for pro-regenerative or protective compounds. Together, the article provides a conceptual and practical overview that will help researchers adopt and further develop this versatile approach in regenerative biology. It will be of interest to researchers studying regeneration, disease modelling, and targeted cell ablation, particularly those working with zebrafish and other genetic model systems.

*For correspondence:
mparson1@uci.edu

**Competing interest:** The authors declare that no competing interests exist.

**Abstract** Here, we review the history, advancements, and broad utility of the NTR/prodrug system, and suggest future strategies for developing versatile ablation models. As a chemo-genetic tool, the nitroreductase (NTR)/prodrug system enables precise spatiotemporal control over cell ablation. The technology leverages bacterial NTR enzymes (e.g. *nfsB*) to convert inert prodrugs into cytotoxic agents, thereby allowing researchers to induce targeted cell death. Although the NTR/prodrug approach was first implemented in transgenic mice, it was subsequently adapted to zebrafish, where it has been extensively optimized and applied. Consequently, zebrafish remain the primary focus of this review. Nevertheless, the utility of the NTR/prodrug system has expanded to other important model organisms, including *Drosophila*, *Nematostella*, *Xenopus*, medaka, and rats, enabling detailed studies of tissue damage and regeneration. This review highlights how the NTR system has been deployed to model a spectrum of human diseases, including Parkinson's disease, retinal degeneration, demyelinating disorders, and kidney disease. These models provide valuable platforms to study pathogenesis in vivo. Furthermore, the precise and controllable nature of NTR ablation makes it an ideal tool for high-throughput chemical and genetic screens aimed at discovering pro-regenerative and protective compounds. The development of NTR2.0, an enzyme variant with over 100-fold greater activity, along with more potent prodrugs such as ronidazole (RNZ), has dramatically broadened experimental possibilities. These improvements permit chronic ablation and long-term disease modeling at well-tolerated drug concentrations. Here, we present some key considerations, including transgenic design for optimal cell-type specificity, calibrating expression levels for desired ablation kinetics, and suitable controls to allow interpretation. These best practices will allow the researcher to develop a precise, reproducible, and versatile platform for either modeling human disease or dissecting regenerative mechanisms.

## Introduction

### Why regeneration?

We often regard ourselves as having limited regenerative capacity, yet numerous human tissues exhibit substantial renewal: the liver restores lost mass, the skin undergoes continuous stem cell-driven turnover, skeletal muscle repairs through satellite cells, bone heals via remodeling, blood is replenished by hematopoietic stem cells, the intestinal epithelium renews within days, and the endometrium regenerates cyclically. However, the regenerative capacity of human tissues is modest when contrasted with that of many other species, in which entire organs or appendages can be restored following injury (*Poss, 2010*; *Tanaka and Reddien, 2011*). This capacity is profound in many invertebrates; for instance, hydra and planarians can regenerate complete organisms from minute fragments (*Pellettieri, 2019*).

Understanding these enviable examples provides a framework for uncovering the cellular and molecular mechanisms that organisms employ to repair or replace damaged tissues, with the ultimate goal of developing new strategies to combat human injuries, degenerative diseases, and age-related decline (*Gurtner et al., 2008*; *Jopling et al., 2011*). An essential requirement for this research is the ability to induce controlled, reproducible injury to study the subsequent repair processes.

The laboratory mouse (*Mus musculus*) is a critical vertebrate model for human biology and has been essential for developing inducible, cell-type-specific genetic tools in mammals. Like adult humans, adult mice exhibit regenerative and reparative abilities across several tissues; however, their capacity for epimorphic regeneration (regrowth of complex structures such as heart tissue or appendages) is largely confined to early postnatal stages. For developing precise ablation tools, a genetically tractable and cost-effective vertebrate with robust adult regeneration is desirable. The zebrafish (*Danio rerio*) is uniquely suited for this role. Its external development, transparency, and high fecundity facilitate direct in vivo visualization and high-throughput screening (*Patton et al., 2021*). Critically, zebrafish exhibit widespread regenerative abilities in adulthood, fully restoring complex organs, including the heart (*Poss et al., 2002*; *Ross Stewart et al., 2022*), retina (*Hammer et al., 2021*; *Montgomery et al., 2010*), spinal cord (*Becker et al., 1997*; *Zhou et al., 2023*), and fins (*Azevedo et al., 2011*; *Sehring et al., 2022*). These attributes make zebrafish a practical system for dissecting the mechanisms of vertebrate regeneration (*Marques et al., 2019*).

## Methods of ablation in zebrafish

Regeneration studies rely on experimental injuries, and the nature of the injury profoundly shapes the ensuing repair response. In zebrafish, the major injury paradigms can be broadly grouped into the following categories:

**1) Physical injury**: Direct tissue damage via surgical intervention that has been used successfully to study adult regeneration, including fin amputation (*Pfefferli and Jaźwińska, 2015*), ventricular resection (*Poss et al., 2002*), brain lesioning (*Kroehne et al., 2011*), scale removal (*Cox et al., 2018*; *Bergen et al., 2022*), and transection of both tendons (*Tsai et al., 2023*) and ligaments (*Anderson et al., 2023*). Severing of the spinal cord predictably leads to paralysis in zebrafish, but as testament to their regenerative capacity, full mobility is returned by 8 weeks (*Becker et al., 1997*; *Reimer et al., 2008*; *Goldshmit et al., 2012*).

Several non-surgical physical methods can be used to induce tissue damage in animal models, with their suitability largely determined by the subject's size and developmental stage. In adult models, approaches include cryoinjury to simulate myocardial infarction (*González-Rosa and Mercader, 2012*; *Bise et al., 2020*), intense light exposure to trigger photoreceptor degeneration in retinal regeneration studies (*Vihtelic and Hyde, 2000*), and acoustic trauma to model inner-ear damage and hearing loss recovery (*Schuck and Smith, 2009*; *Liang et al., 2012*). The acoustic method has also been applied to larval zebrafish to target lateral line hair cells (*Uribe et al., 2018*). Larval and embryonic zebrafish, owing to their small size and optical transparency, permit highly precise laser ablation of single neurons (*Liu and Fetcho, 1999*; *Muto and Kawakami, 2018*; *Roeser and Baier, 2003*) or even individual cardiomyocytes (*Matrone et al., 2013*). Additionally, thermal injury has been employed in larvae to model burns and skin regeneration, revealing rapid inflammatory cell recruitment (*Miskolci et al., 2019*) and keratinocyte migration, but impaired sensory axon regeneration compared to mechanical injury (*Fister et al., 2024*).

**Table 1.** Representative pharmacological toxins used for cell-specific ablation in zebrafish.
These compounds enable dose-controlled injury across diverse tissues but often exhibit limited specificity and off-target effects that can complicate interpretation of regenerative responses.

| Toxin | Target Cell Type | Tissue/Organ | Key Limitation | Ref. |
|---|---|---|---|---|
| Acetaminophen | Hepatocytes | Liver | Dose-dependent toxicity with systemic side effects. | *North et al., 2010* |
| Aminoglycosides (neomycin, gentamicin) | Sensory hair cells | Lateral line, inner ear | Differential effects by compound. Incomplete ablation at some doses. | *Coffin et al., 2013*; *Thomas and Raible, 2019*; *Uribe et al., 2013*; *Wiedenhoft et al., 2017* |
| Caerulein | Acinar cells | Pancreas | Induces pancreatitis and leads to destruction of adjacent tissue | *Falcão et al., 2024*; *Kim, 2008* |
| Cisplatin | Sensory hair cells | Lateral line, inner ear | Damages support cells and delays regeneration. Nephrotoxicity and ototoxicity. | *Lee et al., 2024* |
| Copper sulfate (CuSO$_4$) | Sensory hair cells, support cells | Lateral line | At higher doses also damages support cells and afferent neurons, impairing regeneration. | *Holmgren et al., 2021* |
| 6-Hydroxydopamine (6-OHDA) | Dopaminergic neurons | Brain | Broad catecholaminergic toxicity. May require direct injection in some models. | *Dovonou et al., 2023* |
| Ouabain | Retinal neurons | Retina | Dose-dependent and can damage multiple retinal cell layers. | *Fimbel et al., 2007*; *Sherpa et al., 2008* |
| MoTP | Melanocytes | Skin (pigment system) | Ablation is developmentally restricted | *Yang and Johnson, 2006* |
| MPTP / MPP$^+$ | Dopaminergic neurons | Brain | Species-dependent metabolism. Strict handling required. Off-target effects. | *Dovonou et al., 2023* |
| Streptozotocin (STZ) | Pancreatic β-cells | Pancreas | Off-target effects, e.g. hepatotoxicity | *Moss et al., 2009* |

Regardless of the method, each of these physical injury approaches perturbs tissue integrity in characteristic ways, disrupting multiple components of tissue architecture, including extracellular matrix, mesenchymal organization, and local vasculature. This disruption drives rapid physiological and robust inflammatory responses involving neutrophils and macrophages, all of which critically influence regenerative outcomes (*Miskolci et al., 2019*; *Fister et al., 2024*; *Gelashvili et al., 2026*).

**2) Cell-specific toxins**: While physical injury has provided key insights into regenerative biology, these approaches are often constrained by limited precision and cell-type specificity. Pharmacological methods offer a complementary strategy, using small molecules to induce targeted, dose-controlled damage. *Table 1* summarizes representative compounds used for cell-specific ablation in zebrafish. However, many of these agents introduce off-target effects, ranging from dose-dependent systemic toxicity to unintended tissue injury, which can make downstream regenerative responses harder to interpret. For example, MPTP and MPP$^+$ mainly target dopaminergic (DA) neurons but can also impact noradrenergic and serotonergic systems via uptake through their neurotransmitter transporters (*Dovonou et al., 2023*; *Meredith and Rademacher, 2011*). Similarly, aminoglycoside antibiotics, while commonly used to ablate lateral line hair cells, can also damage support cells and afferent neurons at higher concentrations, and exhibit compound-specific differences in ototoxicity (*Coffin et al., 2013*; *Thomas and Raible, 2019*; *Uribe et al., 2013*; *Wiedenhoft et al., 2017*). Streptozotocin, widely used to ablate pancreatic β-cells, is known to cause broader cytotoxicity beyond the target population (*Moss et al., 2009*). To achieve higher precision and reduce off-target effects, as well as expand the kinds of cells that can be ablated, the field has increasingly turned to genetic strategies. These methods allow damage to be targeted with exquisite specificity to predefined cell types (*Table 1*).

**3) Optogenetic cell ablation**: Due to the transparent nature of young zebrafish, this model is highly amenable to optogenetic techniques that enable precise, non-invasive control of cellular signaling, neuronal circuit activity, and targeted cell ablation (see reviews *Varady and Distel, 2020*; *Baillie et al., 2021*). KillerRed is a genetically encoded photosensitizer that produces reactive oxygen species (ROS) upon green or yellow light illumination, killing nearby cells (*Buckley et al., 2017*; *Bulina et al.,*

*2006*). When expressed under cell-specific promoters, this technique allows researchers to eliminate defined cell populations in living embryos or larvae with precise spatiotemporal control (*Buckley et al., 2017*; *Bulina et al., 2006*; *Teh et al., 2010*). A key advantage of this method is its speed, as KillerRed-expressing cells can be ablated within hours of light exposure. This approach is particularly useful for modeling diseases in which ROS-mediated cell death is central, such as neurodegeneration (*Formella et al., 2018*) and cardiomyopathies (*Teh and Korzh, 2014*), but it has not yet been applied in conventional regeneration studies. Although speculative, this method appears best suited for discrete, optically accessible cells (*Buckley et al., 2017*), and is likely to be impractical for ablation of large populations, whole tissues, or cells deep within the adult body, where localized illumination is more difficult. Because this method relies on controlled optical delivery via microscopy, it also limits throughput, making high-volume studies challenging.

**4) Chemogenetic, cell-specific ablation**: Chemogenetic ablation relies on transgene-driven, cell-specific expression of an exogenous protein that converts an otherwise innocuous chemical into a cytotoxic agent. By eliminating only the intended population and leaving neighboring tissues intact, these methods generate narrowly focused injuries whose regenerative responses differ substantially from those induced by broader, multi-tissue physical damage. Four such chemogenetic systems established in zebrafish include:

i.  Human diphtheria toxin receptor and diphtheria toxin: The human pathogen *Corynebacterium diphtheriae* produces diphtheria toxin (DT), which enters cells by binding the human diphtheria toxin receptor (hDTR) (*Collier, 2001*; *Naglich et al., 1992*). Because endogenous receptors in non-primates do not bind DT efficiently, these animals are naturally resistant to the toxin. This strategy allows researchers to engineer specific cell types to express hDTR, making them uniquely susceptible to DT-induced ablation (*Saito et al., 2001*). In mouse research, the DTR/DT system has become one of the most commonly employed strategies for conditional cell ablation, owing to its rapid and reliable elimination of targeted cell populations across many tissues. However, the method has caveats, like the fact that DT alone can cause kidney damage (*Goldwich et al., 2012*) and that repeated DT exposure induces production of neutralizing antibodies, preventing long-term or chronic ablation.

    Although DTR/DT approaches are widely used in mouse studies, their application in zebrafish has been more limited (*Jimenez et al., 2021*; *Schmitner et al., 2017*). A noteworthy example is the transgenic line generated by Jimenez et al., in which hDTR driven by the hair-cell-specific *myo6b* promoter enabled selective depletion of sensory hair cells following DT administration, with full regeneration occurring within days (*Jimenez et al., 2021*). Some studies in zebrafish have instead expressed diphtheria toxin subunit A (DTA) directly in target cells (*Kurita et al., 2003*; *Li et al., 2009*), although constitutive DTA expression lacks temporal control and is not suitable for regeneration studies. Temporal control can be introduced through Cre/lox-inducible DTA systems (*Sun et al., 2021*), but issues such as recombination efficiency, mosaicism, and promoter leakiness may limit precision (*Erhardt et al., 2025*).

ii. Inducible caspase systems: Caspase cascade in apoptosis begins with upstream initiator caspases (e.g. caspase 8) activated by dimerization, which then activates effector caspases (e.g. caspase 3) to execute cellular dismantling (*Salvesen and Dixit, 1997*; *Boatright and Salvesen, 2003*; *Riedl and Salvesen, 2007*). In zebrafish, researchers have exploited induction of caspase-8 dimerization to achieve temporal and spatial control of cell-specific ablation in two main ways: (1) activation by the FK1012 chemical inducer of dimerization (*Schmitner et al., 2017*; *Banaszynski et al., 2006*) and (2) expression of a fusion between caspase and the modified estrogen receptor ligand-binding domain (ERT2), whose activity is induced by binding of tamoxifen (*Pose-Méndez et al., 2023*; *Weber et al., 2016*; *Chu et al., 2008*). Compared to FK1012, tamoxifen pharmacokinetics are better characterized in zebrafish, and a convincing study showing lifelong regeneration of cerebellar Purkinje cells in zebrafish strongly supports the use of the ER-T2/tamoxifen-inducible ablation approach for probing cell loss and recovery (*Pose-Méndez et al., 2023*).

iii. HSV-TK: The herpes simplex virus thymidine kinase (HSV-tk) has been applied in zebrafish for conditional cell ablation. Transgenic expression of this 'suicide gene' (here defined as a gene encoding an enzyme that converts a nontoxic prodrug into a cytotoxin; *Spooner et al., 2001*) in a defined cell population converts the antiviral prodrug ganciclovir into toxic nucleotides that are lethal to proliferating cells, but this dependence on cell division has limited the utility of the approach (*Moro et al., 2009*).

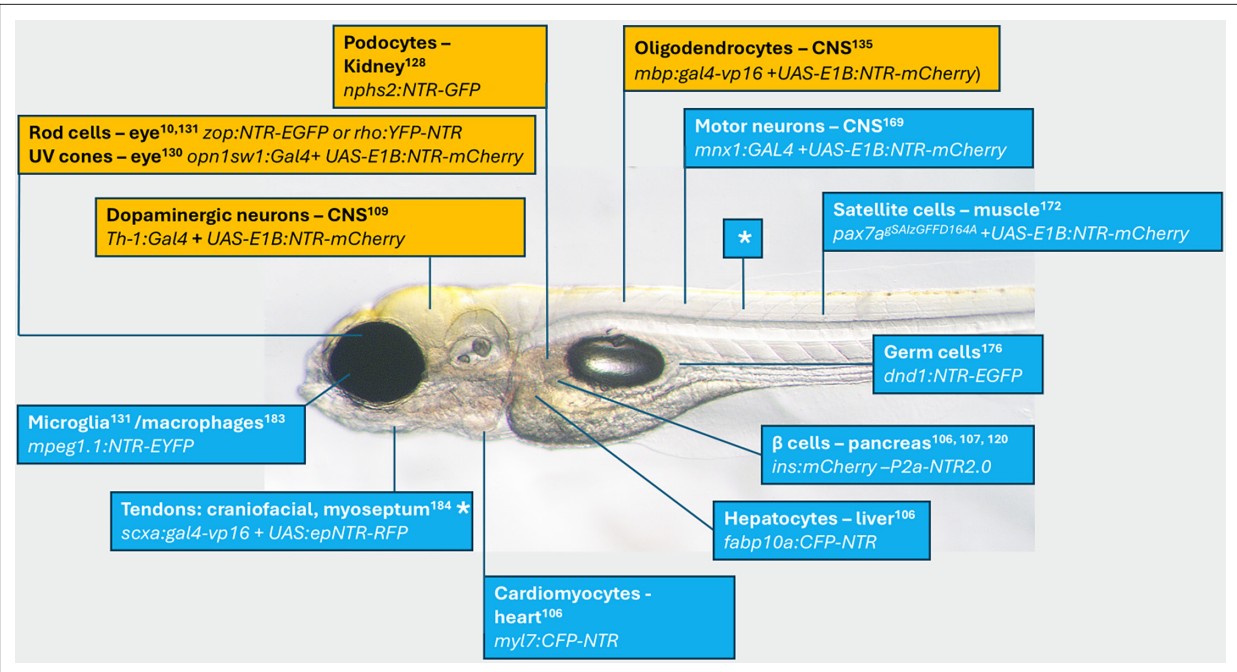

**Figure 1.** Commonly targeted cell types for ablation studies in zebrafish. See *Supplementary file 1* for a more complete list. Image of a 5-day post-fertilization (dpf) casper zebrafish larva (*White et al., 2008*) with approximate position of cell type ablated. Name of cell and transgene provided along with reference. Orange highlights indicate transgenic models used to study human pathologies. *Same line two different tissues (*White et al., 2017*; *Brandt et al., 2021*; *Niu et al., 2020*).

iv. Nitroreductase (NTR): Bacterial nitroreductase genes (nfsB) are another class of suicide genes. These genes encode NTR, enzymes that convert nitro-containing prodrugs (e.g. nitroimidazoles and nitrofurans) into cytotoxic metabolites. As a result, transgenic animal cells expressing NTR become vulnerable to treatment with prodrugs such as metronidazole (MTZ), ronidazole (RNZ), nifurpirinol (NFP), and CB1954. Unlike HSV-TK, the NTR/MTZ system operates as a 'cell-cycle-independent' method for targeted cell ablation, and improved variants of NTR can efficiently ablate fully differentiated cell types (*Sharrock et al., 2022*; *Mulligan and Mumm, 2024*).

The NTR/prodrug system is now a staple of zebrafish research, although, somewhat surprisingly, the approach was first developed in the mouse. Since then, it has been adapted for use in a wide range of model organisms, including:

- *Drosophila melanogaster* – Teeters et al. used NTR and RNZ to ablate multiple, diverse cell types during development, demonstrating rapid, temperature-independent ablation using a simple drug-feeding protocol (*Teeters et al., 2025*).
- *Nematostella* – Gavgani et al. used NTR and NFP to ablate neurons and reveal their requirement in body-axis regeneration (*Mazloumi Gavgani et al., 2025*).
- *Xenopus laevis* – Two NTR/MTZ models, one targeting oligodendrocytes and the other rod photoreceptors, achieved reliable, cell-specific ablation with subsequent regeneration. One targeting study noted temperature-dependent NTR activity, with reduced efficiency below 22°C (*Kaya et al., 2012*; *Langhe et al., 2017*; *Mannioui et al., 2018*; *Martinez-De Luna and Zuber, 2018*).
- Medaka – Willems et al. used NTR and MTZ to conditionally ablate osteoblasts and assess regeneration following drug withdrawal (*Willems et al., 2012*).
- Rat – Kwak et al. applied NTR and CB1954 to ablate neonatal cerebellar and ventricular progenitors, resulting in ataxia and reduced cerebellar volume (*Kwak et al., 2007*).
- Zebrafish – The most extensively used model for NTR/prodrug-mediated cell ablation, with applications across a wide range of tissues to study regeneration and to model human disease (*Figure 1* and *Supplementary file 1*).
- Mouse – The first in vivo NTR/CB1954 transgenic models were generated in mouse, where NTR expression enabled selective ablation of T cells and mammary luminal epithelial cells (*Clark*

*et al., 1997*; *Drabek et al., 1997*). These initial studies were followed by ablation studies of adipocytes, neurons, and kidney podocytes, showing the versatility across diverse tissues of this ablation method (*Felmer et al., 2002*; *Isles et al., 2001*; *Macary et al., 2010*). However, most of these efforts were carried out in the context of evaluating NTR as a suicide-gene strategy for cancer therapy in humans, rather than a tool for studying cell regeneration. Subsequent work in mammals shifted toward the DTR system, which is the predominant method of chemogenetic, cell-specific ablation in mouse.

## Development of NTR as a suicide gene

Bacterial NTRs from *Escherichia coli* (*nfsA* and *nfsB*) were first characterized in the 1970s–80s for their role in reducing nitroaromatics (*Bryant et al., 1981*; *Whiteway et al., 1998*; *McCalla et al., 1978*), a function later harnessed in the 1990s for Directed Enzyme Prodrug Therapy (DEPT) (*Drabek et al., 1997*; *Zawilska et al., 2013*; *Karjoo et al., 2016*). DEPT strategies use viral or antibody-based systems to deliver a 'suicide gene' product to tumors (e.g. NTR) (*Knox et al., 1993*). Once localized, the enzyme converts an administered prodrug into a cytotoxic agent, selectively killing the cancer cells.

The prodrug CB1954 is harmless to human cells but becomes a powerful, DNA-damaging toxin after activation by NTR (*Knox et al., 1988*; *Anlezark et al., 1995*). Even though the NTR/CB1954 approach has been tested in clinical trials (*Palmer et al., 2004*; *Patel et al., 2009*), its effectiveness in mitigating cancers is limited due to low NTR activity and slow prodrug metabolism. Despite these therapeutic shortcomings, this work established NTR/CB1954 as a potent conditional cell-killing strategy. However, this system is subject to a 'bystander effect', where activated metabolites diffuse into and kill neighboring cells. This attribute reduces the precision of targeted ablation and introduces ambiguity when assessing subsequent regeneration.

To overcome these limitations, researchers turned to alternative prodrugs. MTZ, a nitroimidazole antibiotic, emerged as a particularly effective option as it is non-toxic to eukaryotic cells until reduced by bacterial NTR. Unlike CB1954, the activated metabolites of MTZ are short-lived and largely confined within the target cell, minimizing bystander effects (*Bridgewater et al., 1997*; *Curado et al., 2008*; *Sharrock et al., 2021*). This property made the NTR/MTZ system especially well suited for regeneration and developmental studies, which benefit from precise, cell-specific ablation. In zebrafish, NTR/MTZ-mediated ablation consistently induces apoptosis across multiple tissues, as first shown in hepatocytes, cardiomyocytes, and pancreatic β-cells (*Curado et al., 2007*; *Pisharath et al., 2007*). Across tissues, NTR/MTZ ablation elicits stress-linked apoptotic pathways, with hepatocytes showing elevated ROS and DA neurons exhibiting early mitochondrial impairment, both culminating in apoptosis (*Stoddard et al., 2019*; *Kim et al., 2021*). The exact *coup de grâce* will likely vary by cell type, reflecting differences in metabolic activity, mitochondrial content, and intrinsic sensitivity to oxidative or genotoxic stress. Ablation kinetics are similarly context-dependent: NTR expression levels, cell identity, developmental stage, and MTZ dose all influence how rapidly and completely cells are eliminated. Defining how these variables shape NTR/MTZ-induced cytotoxicity across tissues remains an important direction for future work.

## NTR/MTZ in regenerative studies

The NTR/MTZ system for cell-specific ablation was first described in two 2007 studies. Pisharath et al. placed the *E. coli* nfsB gene (NTR) under the zebrafish insulin promoter to express an NTR-mCherry fusion in pancreatic β-cells; treatment with 10 mM MTZ produced complete β-cell loss without affecting neighboring α-cells or exocrine tissue (*Pisharath et al., 2007*). Curado et al. used cell-specific promoters to express CFP-NTR in cardiomyocytes, hepatocytes, and β-cells, likewise demonstrating that MTZ induced highly targeted cell death with no detectable bystander effects (*Curado et al., 2007*). Of note, NTR functions robustly whether fused to the N or C terminus of a fluorescent reporter, facilitating flexible transgene design (*Curado et al., 2008*; *Pisharath and Parsons, 2009*). In both studies, tissues regenerated after MTZ withdrawal, establishing the NTR/MTZ approach as a versatile, specific, inducible, and reversible tool for regeneration studies.

Unlike acute physical injury, NTR-mediated ablation induces apoptotic cell death and predominantly recruits macrophages rather than neutrophils (*Stoddard et al., 2019*). This is an important

consideration, as the type of cell death shapes the regenerative response. In zebrafish, macrophages are not merely phagocytic responders but essential regulators of regeneration: they clear apoptotic debris, modulate the inflammatory milieu, and release factors that support appropriate tissue remodeling (*Petrie et al., 2014*; *Bohaud et al., 2021*).

This controlled context of NTR/MTZ ablation provides a platform for identifying the cells responsible for regeneration, a goal that can be achieved by pairing ablation with lineage tracing. For example, following targeted ablation of pancreatic β-cells, researchers used Cre-based lineage tracing to track the origins of regenerated endocrine cells. These studies revealed that Notch-responsive ductal cells called centroacinar cells act as facultative progenitors that delaminate from the ducts and replenish lost β-cells (*Delaspre et al., 2015*; *Beer et al., 2016*; *Ghaye et al., 2015*). A parallel strategy applied to hepatocyte ablation uncovered an analogous regenerative mechanism in the liver where Notch-responsive biliary epithelial cells (cholangiocytes) delaminate, dedifferentiate, proliferate, and redifferentiate into new hepatocytes (*Choi et al., 2014*; *He et al., 2014*). Combining single-cell RNA sequencing (scRNA-seq) of pancreatic ducts and hepatic ducts after β-cell/hepatocyte ablation has been used to map molecular mechanisms and identify intermediate progenitor states as new β-cells/hepatocytes are formed (*Mi et al., 2023*; *Eski et al., 2025*). This integrated paradigm of targeted ablation, Cre-based lineage tracing, and scRNA-seq provides a systematic framework for dissecting the mechanisms that drive tissue regeneration.

Combining Cre-based lineage tracing with NTR ablation can reveal the origins of regenerated cells. Another way to identify progenitors is to induce regeneration and then use NTR to ablate candidate progenitor populations, assessing whether regeneration is subsequently impaired. The Raible lab used two complementary ablation approaches to pinpoint the cells responsible for regeneration (*Thomas and Raible, 2019*). Neomycin, an aminoglycoside antibiotic, reliably ablates mature hair cells (*Coffin et al., 2013*), which normally regenerate fully. In parallel, the NTR/MTZ system was used to selectively ablate dorsoventral (DV) support cells, which were suspected to act as progenitors. When both the mature hair cells and DV support cells were eliminated, regeneration was dramatically impaired (*Thomas and Raible, 2019*). This dual-ablation strategy demonstrates another way the NTR/prodrug system can be used to identify the cell populations that contribute to regeneration.

## NTR/prodrug-dependent ablation in modeling human disease

The NTR/prodrug ablation system also lends itself well to modeling human diseases that are characterized by the specific and progressive loss of distinct cell populations. The core strengths of this chemogenetic approach include cell-type specificity and temporal control, which allows it to be a toolkit for recapitulating pathological events in vivo. This allows for the real-time dissection of disease initiation, progression, and complex cellular responses to injury. A diverse array of human pathologies has been modeled using NTR/MTZ in zebrafish, including chronic hyperglycemia (a symptom of diabetes) (*Tucker et al., 2023*), acute liver damage (*Choi et al., 2014*), and cardiac injury (*Curado et al., 2007*; *Palencia-Desai et al., 2015*; *Apolínová et al., 2024*). Here, we analyze selected models in greater detail, focusing on kidney disease, retinal degeneration, demyelinating disorders, and neurodegeneration (*Figure 1* – orange boxes).

### Kidney glomerular disease

Glomerular diseases stem from a common problem: progressive podocyte loss or dysfunction. These cells are crucial for maintaining the kidney's filtration barrier, and their damage leads to proteinuria, the leakage of abnormal amounts of protein into the urine (*Greka and Mundel, 2012*). To model this pathology, a podocyte-specific NTR mouse line was generated, in which NTR expression under the *podocin* promoter enables inducible podocyte injury (*Macary et al., 2010*). Administration of CB1954 triggered acute podocyte damage with proteinuria and leads to progressive focal segmental glomerulosclerosis, providing a mammalian proof-of-principle for NTR/CB1954-mediated podocyte ablation, though this approach was never widely adopted. Subsequent mouse studies of podocyte ablation instead used DTR-based approaches (*Wharram et al., 2005*; *Zhou et al., 2015*; *Stevens et al., 2018*), even though DT alone can induce transient podocyte injury and proteinuria in wild-type mice (*Goldwich et al., 2012*).

Other investigators employed the NTR/MTZ to induce podocyte-specific loss in zebrafish. Researchers used the *nphs2* promoter to drive NTR in transgenic zebrafish (nphs2:NTR-GFP), enabling precise, inducible podocyte ablation (*Zhou and Hildebrandt, 2012*; *Huang et al., 2013*).

Administering MTZ triggered rapid podocyte apoptosis, resulting in classic features of human glomerular injury, including disruption of the filtration barrier, proteinuria, and edema. By mirroring these essential aspects, this model provides a direct and relevant system for studying the progression of human podocytopathies. Furthermore, the zebrafish pronephros allows live imaging of podocyte injury and subsequent regeneration (*Zhou and Hildebrandt, 2012*). After MTZ withdrawal, podocyte repopulation occurs through residual cells and local progenitors (*Huang et al., 2013*). This makes the model an ideal platform for studying podocyte repair and uncovering pathways with therapeutic relevance.

## Retinal degeneration

Given the high conservation of eye structure between zebrafish and humans, the NTR/MTZ system provides an excellent platform for modeling inherited retinal degenerations such as retinitis pigmentosa and cone dystrophies, conditions in which progressive photoreceptor loss leads to vision decline (*Narayan et al., 2016*). By driving NTR expression under photoreceptor-specific promoters, distinct photoreceptor subtypes can be selectively ablated. For example, expression under the rhodopsin (rho) promoter enables targeted elimination of rod photoreceptors, providing a robust zebrafish model of retinitis pigmentosa (*Montgomery et al., 2010*). Similarly, the use of cone opsin promoters such as *opn1sw1* permits ablation of defined cone populations to study cone dystrophies (*Fraser et al., 2013*). This targeted ablation triggers apoptotic photoreceptor loss while sparing neighboring retinal cells (*White et al., 2017*). A major advantage of the zebrafish system is its capacity for spontaneous retinal regeneration, driven by the dedifferentiation and proliferation of Müller glia that give rise to new photoreceptors (*Bernardos et al., 2007*; *Fausett and Goldman, 2006*). The NTR/MTZ paradigm allows for precise initiation and synchronization of this regenerative process, enabling real-time dissection of the cellular and molecular programs underlying photoreceptor replacement and the contributions of innate immune signaling to retinal repair (*Langhe et al., 2017*).

## Demyelinating disorders

Conventional autoimmune models of multiple sclerosis (MS), including experimental autoimmune encephalomyelitis (EAE), often display substantial variability in the timing and severity of disease onset, alongside a highly complex and multifactorial immunopathology (*Constantinescu et al., 2011*). This inherent heterogeneity makes it difficult to disentangle the individual cellular and molecular events that specifically contribute to successful remyelination, thereby limiting the ability to clearly define the mechanisms required for effective tissue repair. To overcome these hurdles, NTR-MTZ-based models have been utilized to ablate oligodendrocytes and their progenitors. Tg(*mbp:-gal4-vp16*); Tg(*UAS-E1B:NTR-mCherry*) fish express NTR specifically in mature oligodendrocytes that myelinate CNS axons. Exposure to MTZ in these fish caused rapid and synchronized demyelination within 48 hr, characterized by the retraction of myelin sheaths and oligodendrocyte cell death (*Chung et al., 2013*). Furthermore, subsequent regeneration resulted in myelin sheaths that restored normal length and thickness correlated to axon caliber (*Karttunen et al., 2017*). This mechanistic parallel is highly relevant, as the failure to restore proper myelin architecture is a central hallmark of progressive disability in human demyelinating diseases (*Constantinescu et al., 2011*).

## DA neurodegeneration

Traditional genetic models of Parkinson's disease (PD) often exhibit weak or late-onset phenotypes. Neurotoxin-based models frequently induce off-target neuronal loss, and the compounds themselves pose safety risks to researchers, restricting their use in scalable or high-content screening applications (*Dovonou et al., 2023*; *Meredith and Rademacher, 2011*). To overcome these hurdles, Kim et al. utilized a chemogenetic model in zebrafish to ablate DA neurons (*Kim et al., 2021*). NTR expression was driven from the *tyrosine hydroxylase* (*th*) promoter (the *th1* gene encodes an enzyme required for dopamine synthesis). *Th:NTR* fish express NTR1.0 in the DA neurons of the ventral forebrain, the zebrafish homolog of the mammalian substantia nigra.

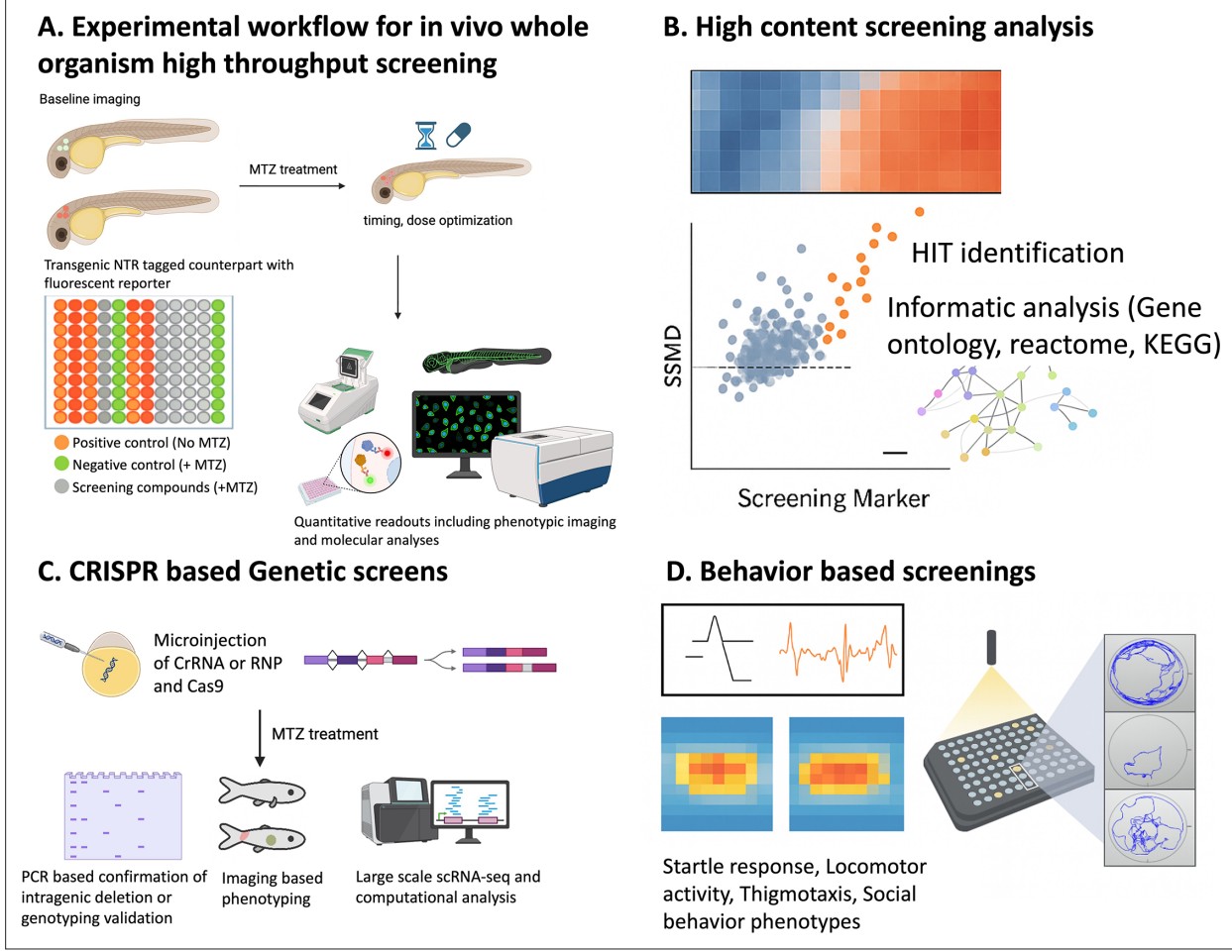

**Figure 2.** Nitroreductase (NTR)/metronidazole (MTZ)-based screening platforms in zebrafish. Overview of the integrated chemogenetic screening workflow using NTR-mediated ablation. (**A**) Experimental design showing transgenic zebrafish expressing NTR in target tissues, baseline imaging, and subsequent MTZ treatment to induce cell-type-specific ablation. The use of parallel transgenic controls and multiwell plate layout enables quantitative assessment of tissue loss and recovery. (**B**) High-content chemical screening pipeline integrating automated imaging, hit identification, and pathway-level analysis using standardized statistical metrics. (**C**) Genetic screening framework coupling sgRNA-based mutagenesis with imaging-based phenotype scoring to uncover modifiers of cell loss or regeneration. (**D**) Behavioral assays to quantify functional recovery or pharmacological response.

Exposure to MTZ in *th:NTR* fish caused pronounced mitochondrial damage within DA neurons, including mtDNA damage, impaired mitochondrial function, reduced organelle motility, and altered morphology, ultimately resulting in neuron loss (*Kim et al., 2021*). The finding that NTR/MTZ ablation kills DA neurons through mitochondrial dysfunction is particularly significant, as mitochondrial impairment is a central pathological hallmark of human PD (*Henrich et al., 2023*). By recapitulating this key feature, the *th:NTR* model moves beyond a simple cell-elimination system to one with strong disease relevance, providing a robust, scalable, and experimentally tractable platform ideally suited for screening small molecules that protect DA neurons or modulate PD-associated pathways. Furthermore, this work remains one of the few detailed mechanistic investigations of NTR/MTZ-mediated cytotoxicity in a defined zebrafish neuronal population, indicating that mitochondrial injury, rather than early nuclear DNA damage, plays a major role in driving NTR/MTZ-induced DA neuron death.

## NTR/prodrug-based screening

The NTR/MTZ ablation system provides a reproducible and scalable platform for functional screening in zebrafish, combining cell-type specificity, quantitative imaging, and compatibility with both chemical and genetic perturbations (*Figure 2*).

## Small-molecule screening

The NTR/MTZ ablation system has been adapted for high-content chemical screening in zebrafish, allowing quantitative evaluation of compound effects on cell death, protection, and regeneration across diverse tissues. In the context of retinal degeneration (*Zhang et al., 2021*), Mumm and colleagues demonstrated the high-throughput capabilities of the system by screening 2934 compounds using the *Tg(rho:YFP-NTR)* model of retinitis pigmentosa. By driving NTR specifically in rod photoreceptors, the lab induced targeted cell death and screened for small molecules that could preserve YFP-positive cells despite MTZ exposure. This large-scale effort identified 11 validated neuroprotectants (distinct from simple antioxidants) that were subsequently shown to have conserved efficacy in mouse retinal explant assays (*Zhang et al., 2021*). This cross-species validation confirms that the zebrafish NTR system effectively filters for compounds with relevant translational potential for human blindness.

*Kim et al., 2021* utilized the *Tg(th:NTR)* model to perform a 1403-compound screen for PD. By integrating automated imaging with rigorous statistical metrics, including the Brain Health Score (BHS) and the Strictly Standardized Mean Difference (SSMD), the researchers identified 57 compounds that preserved DA neurons (*Figure 2A and B*). Importantly, the study advanced beyond simple measurements of cell survival and provided mechanistic validation that these compounds protected neurons by restoring mitochondrial function, which is a central hallmark of PD pathology. The predictive validity of these hits was further confirmed through cross-assay validation in a separate Gaucher disease behavior model, demonstrating the system's capacity to identify robust therapeutics for complex neurodegenerative conditions (*Figure 2D*, *Kim et al., 2021*).

Another promising application of the NTR system lies in the identification of therapeutic agents that actively promote tissue regeneration. *Lee et al., 2025*, leveraged an optimized QF-based binary expression system (*mbpa:qf2;quas:epNTR-P2A-mCherry*) to perform a remyelination phenotypic screen for regenerative compounds. This transgenic line achieved greater than 85% oligodendrocyte loss following treatment with 2 mM MTZ for 18 hr, creating a highly reproducible regenerative baseline. Using this platform to screen a kinase-inhibitor library, the authors identified the TGF-β receptor I inhibitor AZ-12601011 as a potent driver of remyelination (*Lee et al., 2025*). Mechanistic validation revealed that this compound promotes repair by modulating microglial and progenitor activation, thereby confirming the system's predictive validity for discovering clinically relevant restorative therapeutics that actively drive the reconstruction of functional tissue.

Similar regenerative screens have been successfully implemented in other tissues, such as the pancreas. *Andersson et al., 2012*, utilized the *Tg(ins:CFP-NTR)* line, crossed with a *Tg(ins:Kaede)* reporter to induce complete β-cell ablation and then monitor the formation of new β-cells. This model was used in a high-content screen of approximately 7000 small molecules to find compounds that would enhance regeneration of the insulin-producing β-cells (*Andersson et al., 2012*). This screen identified adenosine receptor agonists, specifically NECA, as potent stimulators of endocrine regeneration. Detailed mechanistic characterization revealed that NECA signals via the A2aa receptor to specifically enhance the proliferation of regenerating β-cells rather than neogenesis, a therapeutic pathway that was subsequently validated to restore normoglycemia in a streptozotocin-induced diabetic mouse model (*Andersson et al., 2012*).

## Genetic and CRISPR-based screening

Chemical screens can identify potential therapeutic reagents, though their molecular targets often remain unknown. A complementary approach is to perform reverse-genetic screens that integrate NTR-mediated ablation with CRISPR mutagenesis to identify genes affecting regeneration (*Unal Eroglu et al., 2018*). This mutagenesis is achieved by injecting Cas9 ribonucleoprotein (RNP) complexes multiplexed with several guide RNAs per target gene directly into NTR-transgenic embryos (*Figure 2C*). This F0 'crispant' strategy generates high-efficiency somatic mutations in the first generation (*Wu et al., 2018*), allowing researchers to induce cell-specific ablation with MTZ and immediately quantify the effect of gene disruption on regeneration without the delay of establishing stable mutant lines.

To identify regulators of retinal pigment epithelium (RPE) repair, *Lu et al., 2023*, conducted a focused F0 CRISPR screen targeting 27 candidate genes in rpe65a:nfsB-eGFP larvae (*Lu et al., 2023*). By injecting RNP complexes containing three highly mutagenic guide RNAs per gene, they achieved high-efficiency somatic mutagenesis in F0 injected fish. The NTR/MTZ system induced the synchronized, widespread degeneration of the RPE, which subsequently triggered the secondary loss of

photoreceptors. This screen identified numerous regulators of regeneration and revealed a novel mechanism that regulates the infiltration of phagocytic cells required for clearance of debris and complete regeneration (*Lu et al., 2023*).

To find regulators of retinal ganglion cell (RGC) regeneration, *Emmerich et al., 2024*, performed a large-scale CRISPR screen on 100 genes. Using the isl2b:Gal4; UAS:YFP-NTR2.0 line for RGC ablation, they identified 18 effector genes comprising key transcription factors and signaling pathway components (*Emmerich et al., 2024*). The screen revealed that inhibition of Ascl1a accelerated the regeneration of new RGC neurons.

Finally, the integration of F0 mutagenesis with automated imaging establishes a scalable framework for future genetic screens. The 'ZebraReg' platform utilizes a dual-transgenic line (tbx5a:CreERT2; myh7l:loxP-tagBFP-STOP-loxP-mCherry-NTR) that restricts NTR expression specifically to the heart ventricle (*Apolínová et al., 2024*). Treatment with MTZ ablated approximately 97% of cardiomyocytes, triggering a robust regenerative response that typically restores the tissue within 3 days. By combining this precise injury model with F0 CRISPR mutagenesis followed by immediate phenotyping, the study demonstrates a proof-of-concept workflow to understand the genetic mechanisms of cardiac repair.

## Caveats and improvements to the NTR/MTZ system

The NTR/MTZ system is widely used for diverse applications, but its performance has varied between labs. Key issues include batch-to-batch and preparation variability of MTZ, the need for high MTZ doses (≈10 mM) that can cause off-target toxicity (e.g. developing brain [*Lai et al., 2021*], larval/adult intestine [*Tucker et al., 2023*]), and differential susceptibility of some cell types to ablation (*Pisharath and Parsons, 2009*). Because ablation rate depends on both NTR activity and MTZ dose, researchers have pursued three complementary strategies to improve reproducibility and experimental interpretation: (1) increase NTR expression, (2) engineer higher-activity NTR mutants, and (3) identify more efficacious prodrugs that achieve effective killing at lower, less toxic concentrations. These iterative improvements are aimed at mitigating previous limitations and expanding the range of feasible experimental paradigms and are discussed next.

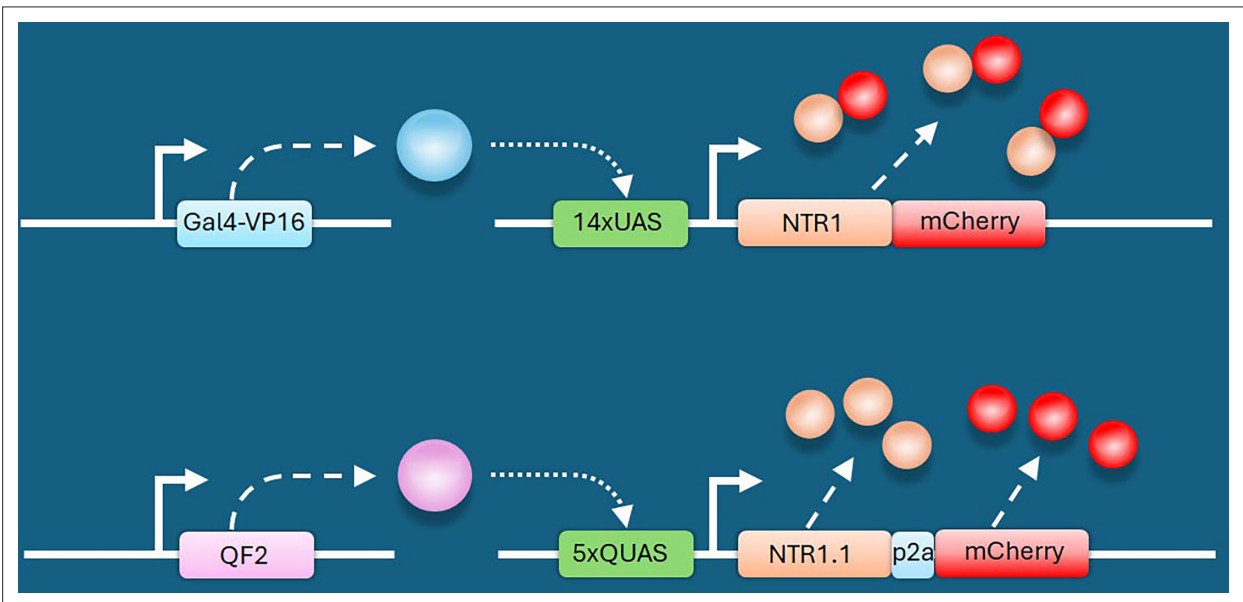

**Figure 3.** Schematic of bipartite systems to drive robust levels of nitroreductase (NTR). On left, driver lines express (dashed arrows) transactivators (**A**) Gal4 (blue sphere) or (**B**) QF2 (purple sphere) under the control of a *cis*-regulatory element (CRE). These transactivators bind their respective upstream activating sequences (either UAS or QUAS, gray boxes) to achieve controlled and amplified NTR expression (tan spheres) in target cells. NTR expression can be monitored by co-production of mCherry (red spheres) either as a fusion protein or as separate proteins due to P2A-dependent ribosome 'skipping' (*Provost et al., 2007*). (**A**) Redrawn from Pisharath and Parsons where the CRE was from *ptf1a* (*Pisharath and Parsons, 2009*) and (**B**) redrawn from Lee et al. where the CRE came from *mbpa* (*Lee et al., 2025*).

1. Increase NTR expression: Strong, well-characterized promoters/enhancers (e.g. the zebrafish insulin promoter) (*Pisharath et al., 2007*) can drive high NTR expression, especially when present in multiple copies via Tol2-mediated transgenesis (*Kalvaitytė et al., 2024*). However, many cell-specific regulatory elements are weak, and maintaining multiple insertions is challenging and prone to genetic drift and intergenerational variability. An alternative is to use a bipartite system such as Gal4/UAS (*Brand and Perrimon, 1993*; *Köster and Fraser, 2001*), which can produce robust, amplified NTR expression even from a single genomic insertion. With this approach, a cell-specific promoter drives a Gal4 transactivator that binds UAS sites to strongly activate *NTR* transcription (*Figure 3*). For example, elements from the 14xUAS constructs of *Köster and Fraser, 2001*, were used to generate the transgenic line Tg(UAS-E1B:NTR-mCherry)[c264] (*Pisharath and Parsons, 2009*; *Davison et al., 2007*). These fish were distributed by the Zebrafish International Resource Center (ZIRC) and have been widely used by the zebrafish community: 10 of the 32 most-cited papers on zebrafish NTR ablation use this line (*Supplementary file 1*). A caveat is that Gal4/UAS DNA elements can be prone to epigenetic silencing, producing mosaic expression; the repetitive UAS contains multiple CpG sites susceptible to DNA methylation (*Halpern et al., 2008*; *Goll et al., 2009*). Silencing can be mitigated by using a less repetitive UAS (e.g. 4×) (*Akitake et al., 2011*) or by using the QF/QUAS bipartite system (derived from Neurospora) (*Potter et al., 2010*), which has been reported to show reduced silencing (*Subedi et al., 2014*) and has recently been adapted for NTR-based ablation (*Figure 3*, *Lee et al., 2025*; *Lengyel et al., 2025*).

2. Higher-activity NTR mutants: Substantial effort has gone into engineering more active NTRs; first driven by their promise as cancer 'suicide-gene' therapies (*Guise et al., 2007*) and later to improve NTR-based ablation in basic research (*Mathias et al., 2014*). Two research groups independently engineered the same three substitutions into the wild-type *E. coli* enzyme (now termed NTR1.0), creating more efficient versions they named epNTR and NTR1.1 (*Guise et al., 2007*; *Mathias et al., 2014*; *Tabor et al., 2014*). Cross-species screening identified a highly active NTR from *Vibrio vulnificus*. Using this enzyme as a scaffold, rational engineering yielded the second-generation variant NTR2.0, which exhibits a greater than 100-fold enhancement in activity over the original NTR1.0 (*Sharrock et al., 2022*).

   ○ The use of first-generation nitroreductase (NTR1) for chronic cell ablation was problematic, as the required 10 mM MTZ dose induces intestinal pathology and approaches the LD50 in zebrafish (*Tucker et al., 2023*). However, the more active NTR2.0 variant enables effective ablation with far lower, better-tolerated MTZ concentrations. To demonstrate this, Tucker et al. developed a zebrafish model expressing NTR2.0 specifically in pancreatic β-cells (*Akitake et al., 2011*). They found that efficient larval β-cell ablation required only 100 µM MTZ, a regimen that could be maintained for 10 days without ill effects. In stark contrast, the NTR1 system required a toxic 10 mM MTZ dose, which is lethal to larvae (independent of NTR) within 3 days. In adult fish, a regimen of 5 mM MTZ for 2 days followed by 2 weeks at 1 mM was completely tolerated by wild-type fish with no ill effects but induced sustained hyperglycemia and weight loss in NTR2.0-expressing fish. This established a powerful model for studying chronic diabetic consequences, such as retinopathy, nephropathy, and impaired wound healing.

   ○ This well-tolerated ablation paradigm now makes it possible to model a range of other chronic conditions, including neurodegenerative, renal, and muscular disorders. This capability, in turn, facilitates the study of long-term disease progression and the evaluation of new therapeutic interventions.

3. More efficacious prodrugs: MTZ efficacy can vary across suppliers and batches. To ensure consistency, it is recommended to prepare fresh MTZ solutions for experiments (*Pisharath and Parsons, 2009*; *Tucker et al., 2023*). To overcome MTZ's limitations, alternative prodrugs like NFP have been tested. NFP is a more potent nitrofuran-based prodrug (*Pisharath and Parsons, 2009*; *Tucker et al., 2023*). However, its structural class is distinct from the nitroimidazole-based prodrugs for which NTR2.0 was specifically engineered (*Sharrock et al., 2022*). As a nitroimidazole prodrug, RNZ likely retains compatibility with newer NTR systems while offering significant practical advantages over MTZ, primarily its better potency (*Teeters et al., 2025*; *Lai et al., 2021*; *Chen et al., 2023*). For instance, in Tg(*fabp10:mCherry-NTR1*) fish, 2 mM RNZ achieved hepatocyte ablation comparable to 10 mM MTZ, a fivefold increase in potency (*Chen et al., 2023*). This pattern of higher efficacy was replicated in a macrophage model, where a fivefold lower RNZ dose was as effective as MTZ (*Lai et al., 2021*). Lai et al. also reported no bystander effects and demonstrated RNZ efficacy with the NTR1.1 variant. Recently, it has also been shown that RNZ functions with NTR2.0 to cause cell-specific ablation (*Duan et al., 2025*), although a

direct comparison of RNZ versus MTZ with NTR2.0 has not been reported. Whether RNZ shows batch-to-batch variability similar to MTZ has not yet been reported. Given the potential for variability, it would be prudent for researchers to titrate each new batch of RNZ or, alternatively, adopt a dosing strategy that exceeds the minimum effective concentration to ensure consistent ablation results. Nonetheless, it is anticipated that the NTR2.0/RNZ combination will further lower the required prodrug concentrations and minimize off-target activity.

Given this evolving landscape of prodrugs and enzymes, what are the critical factors a researcher must weigh when designing an NTR ablation experiment?

## Experimental design: practical and technical considerations

A successful ablation experiment using the NTR/prodrug system requires careful consideration of three key components: (1) transgenic strategy, (2) optimal NTR activity, and (3) appropriate controls. The optimal design depends on the biological question.

- For regeneration studies, aim for complete ablation to clearly assess neogenesis.
- For functional studies, partial ablation may be sufficient to reveal a phenotype.

1. Transgenic strategy:
   - Regulatory elements: Select regulatory elements that ensure precise tissue or cell-type specificity.
     - Discrete regulation: When a single promoter is insufficient to achieve the desired tissue specificity, use intersectional approaches (e.g. Cre/lox) to restrict expression (*Zhong et al., 2019*).
   - Fluorescent marker: Include an independent fluorescent marker (fusion or 2A reporter, *Provost et al., 2007*) to identify transgenic animals and confirm appropriate expression.
   - Positional effects: NTR transgenes, like any transgene, can exhibit positional effects (leakiness, mosaicism). To ensure reliable lines:
     - Screen multiple founders: Identify ≥5 F0 founders and ideally establish 5 independent F1 lines.
     - Compare stable F1 lines: Confirm that fluorescent-marker expression matches expected regulatory-element activity.
     - Prioritize F1 lines based on:
       - Mendelian transmission, indicating a single-site insertion.
       - Consistent, non-mosaic expression of the fluorescent marker, indicating uniform NTR expression in all intended cells.
       - Robust and reproducible expression, independent of whether the transgene is inherited maternally or paternally.
       - Reliable and consistent ablation of target cell.
2. NTR activity:
   - General principle: Strong NTR expression generally yields faster and more complete ablation (*Sharrock et al., 2022*; *Tucker et al., 2023*).
   - NTR variant: NTR2.0 is currently the most active NTR variant used in zebrafish and is recommended for future studies.
   - Prodrug choice: For most applications, RNZ is the recommended prodrug to start with, due to its higher efficacy and lower required dosing relative to MTZ, which can improve both ablation efficiency and experimental consistency.
   - Enhancing expression: If the promoter driving NTR2.0 is weak, amplify expression using binary systems such as Gal4/UAS or QF/QUAS (*Figure 3*, *Lee et al., 2025*; *Davison et al., 2007*; *Lengyel et al., 2025*).
3. Controls:
   - Validating cell death: To properly interpret ablation results, it is important to confirm that cell death occurred. Useful readouts include:
     - Apoptosis assays: TUNEL, cleaved caspase 3 immunostaining (*Curado et al., 2008*; *Pisharath and Parsons, 2009*).

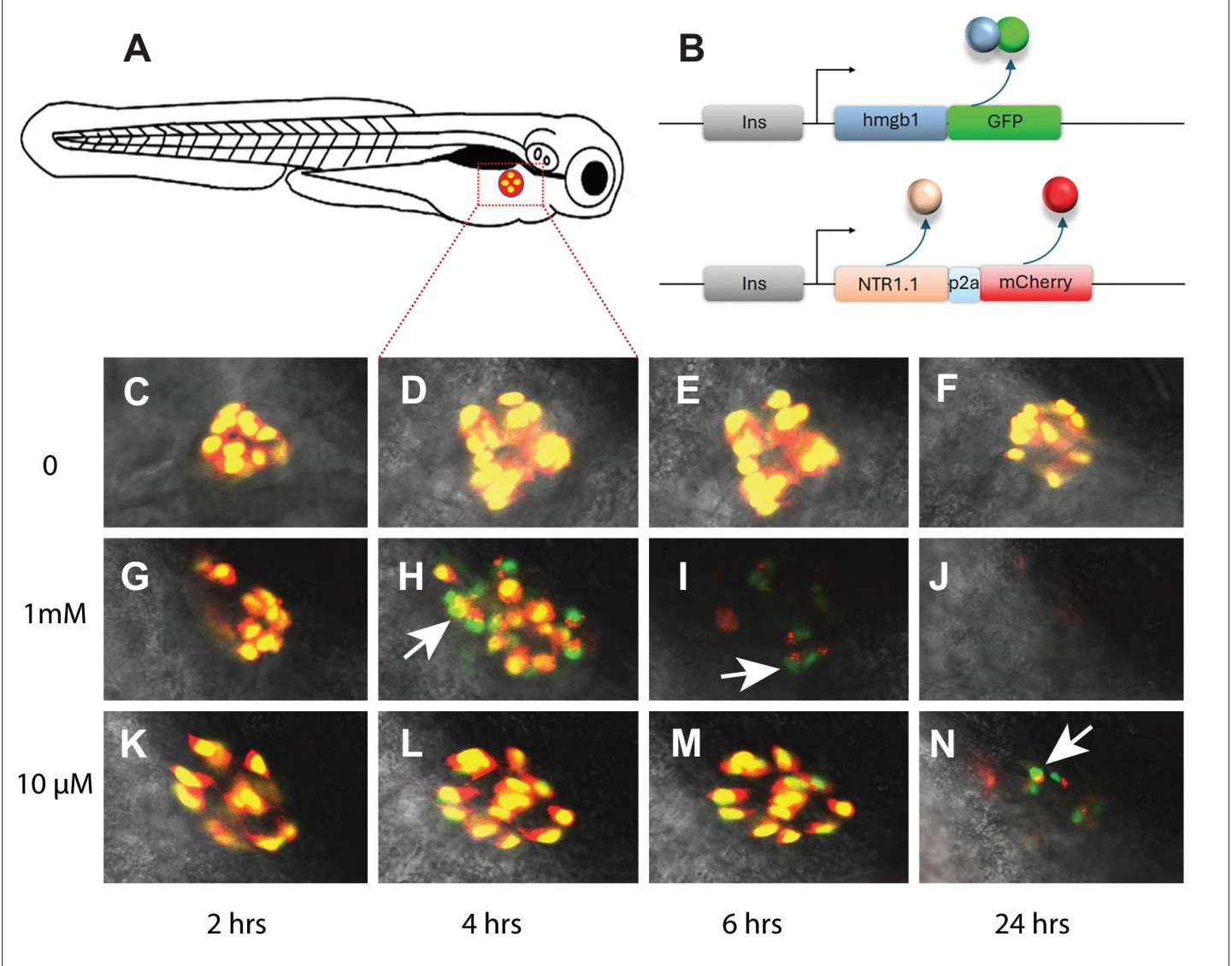

**Figure 4.** Live imaging of cell-death kinetics. (**A**) Schematic of 6-day post-fertilization (dpf) larvae showing position of the pancreatic islet imaged in C-N (red/yellow). (**B**) Diagram of the two transgenes (i*ns:Hmgb1-GFP*, *ins:mCherry-2a-NTR2.0*) in the fish. The insulin promoter (gray box) drives expression of the following: (**B**, above) an Hmgb1-GFP fusion protein and (**B**, below) NTR2.0 and mCherry (presence of the P2A [blue box] makes separate proteins). (**C–N**) Confocal images of the islet in three larval fish over a time course from 6 dpf to 7 dpf (times along the X axis). (**C–F**) Negative control – no MTZ (0). (**G–J**) Fish treated with high MTZ dose (1 mM). (**K–N**) Fish treated with a low MTZ dose (10 µM). (**G–N**) Dying β-cells first lose red fluorescence, revealing green nuclei (arrow heads). A higher dose shows the appearance of green nuclei (**H**) earlier than the lower dose (**N**). (**J**) 24 hr in 1 mM and no debris remains.

- Loss of fluorescent reporters (*Curado et al., 2007*; *Pisharath et al., 2007*; *Pisharath and Parsons, 2009*): If signal perdurance is a concern, use destabilized reporters to reduce fluorescence longevity (*Walker et al., 2012*; *Wang et al., 2015*).
  ○ Cell-death kinetics: As stressed cells may downregulate NTR and escape ablation, validating cell-death kinetics can be informative.
    - Use endpoint analysis (serial time-point fixation + apoptosis markers).
    - Or use cell-death biosensors, such as Hmgb1-GFP, which distinguishes necrosis (nuclear release) from apoptosis (nuclear retention) (*Raucci et al., 2007*; *Scaffidi et al., 2002*).
    - Example: In larvae co-expressing ins:mCherry-2A-NTR2.0 and *ins:hmgb1-eGFP* (*Tucker et al., 2023*; *Parsons et al., 2009*), high-dose MTZ (1 mM) induced apoptosis within

4 hr with complete β-cell loss by 24 hr; a low dose (10 μM) produced slower dynamics (*Figure 4*).
- ○ Negative controls
  - • NTR transgene, no prodrug: Controls for effects of exogenous NTR expression.
  - • No NTR transgene, prodrug: Controls for nonspecific MTZ/RNZ effects, including antimicrobial activity. For microbiome-associated studies, consider alternative ablation methods.

## Additional information

### Funding

| Funder | Grant reference number | Author |
|---|---|---|
| National Institute of Diabetes and Digestive and Kidney Diseases | DK080730 | Michael Parsons |
| National Institute on Aging | NIA AG066519 | Gha-Hyun J Kim |
| UCI Alzheimer's Disease Research Center | | Gha-Hyun J Kim |

The funders had no role in study design, data collection and interpretation, or the decision to submit the work for publication.

### Author contributions

Gha-Hyun J Kim, Resources, Formal analysis, Investigation, Methodology, Writing – original draft, Writing – review and editing; Michael Parsons, Conceptualization, Resources, Data curation, Formal analysis, Supervision, Funding acquisition, Validation, Investigation, Visualization, Methodology, Writing – original draft, Project administration, Writing – review and editing

### Author ORCIDs

Gha-Hyun J Kim (ID) https://orcid.org/0000-0001-9073-2034
Michael Parsons (ID) https://orcid.org/0000-0002-4046-1659

Reviewer #1 (Public review): https://doi.org/10.7554/eLife.110593.3.sa1
Reviewer #2 (Public review): https://doi.org/10.7554/eLife.110593.3.sa2
Reviewer #3 (Public review): https://doi.org/10.7554/eLife.110593.3.sa3
Author response https://doi.org/10.7554/eLife.110593.3.sa4

## Additional files

### Supplementary files

Supplementary file 1. Most-cited publications using the nitroreductase (NTR)/prodrug system. This table lists the most highly cited publications employing the NTR/prodrug system for targeted cell ablation. Candidate studies were identified through a Web of Science search using the query 'nitroreductase ablation', and results were manually curated to include only those papers that directly used an NTR-expressing transgenic line or construct together with a prodrug to induce selective cell death. Entries are ranked by citation count at the time of data collection. For each study, the table reports the targeted cell type or tissue, the species and specific NTR transgenic line used, and the primary biological purpose addressed. The columns ad. (adult) and la. (larvae) indicate whether the transgenic system was used in adults, larvae, or both. The transgene shown in green corresponds to a widely used UAS line that drives Gal4-dependent expression of NTR1 and is included here to aid identification of experiments utilizing this common effector line.

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
