## [Editor Report · eLife Assessment]

This Review Article nicely synthesizes the development, applications, and recent technical advances of the nitroreductase/prodrug system, highlighting how it enables precise spatiotemporal cell ablation and experimental platforms for studying regenerative mechanisms and screening for pro-regenerative or protective compounds. Together, the article provides a conceptual and practical overview that will help researchers adopt and further develop this versatile approach in regenerative biology. It will be of interest to researchers studying regeneration, disease modelling, and targeted cell ablation, particularly those working with zebrafish and other genetic model systems.

---

## [Referee Report · Reviewer #1 (Public review)]

Summary:

Kim and Parsons present a timely overview of the NTR/prodrug system and its applications in regenerative biology research, with particular emphasis on tissue-specific cell ablation. The system has substantially advanced the field by enabling non-invasive, conditional cell elimination, and has proven especially powerful in zebrafish, though applications in other classical model organisms are also noted. The review covers the historical origins of the NTR system, its use in regeneration studies, small-molecule screening, and genetic and CRISPR-based screening, as well as future directions including the development of the highly efficient NTR2 enzyme variant.

Strengths:

This is a useful and well-structured contribution. The manuscript is a valuable resource for the regeneration biology community.

Weaknesses:

The revised manuscript shows significant improvements; however, two points remain insufficiently addressed and should be resolved in the final version.

(1) The term 'suicide gene'

As noted in my first round of revisions, the term 'suicide gene' as applied to bacterial nitroreductase remains unaddressed in the revised manuscript, despite being scientifically inappropriate and a potential source of confusion regarding the NTR/Mtz mechanism.

'Suicide' implies an intrinsic, cell-autonomous programme of self-destruction. This is incompatible with the NTR/Mtz system, in which cell death is experimentally induced through exogenous administration of metronidazole (Mtz) by the investigator. While the 'suicide gene' framing may have utility in the cancer therapy literature, likely to aid communication with non-specialist and clinical audiences, however, it is not standard in the zebrafish field, where NTR is more accurately described as a conditional toxigene. Since this review focuses predominantly on zebrafish models, its terminology should reflect that of the relevant literature.

A further conceptual problem with the 'suicide gene' framing is that it obscures the pharmacological nature of Metronidazole. Mtz is a pharmaceutical agent with intrinsic baseline toxicity: extended exposure or modestly elevated concentrations cause toxic side effects and lethality even in non-transgenic (wild-type) zebrafish (PMID: 24428354). NTR-expressing cells do not self-destruct; rather, they are rendered selectively hypersensitive to Mtz relative to other eukaryotic cells by virtue of expressing the enzyme. This distinction is mechanistically important and should be reflected in the language used throughout the manuscript.

In summary, the term 'suicide gene' does not accurately capture enzyme-mediated bioactivation of an exogenous prodrug and should be removed from the manuscript.

(2) Barriers to using the NTR/Mtz system in non-aquatic model organisms

In response to my suggestion that the title should include "zebrafish" to accurately convey the scope of the review to prospective readers, the authors stated that "there is no intrinsic barrier to adopting this technique more broadly in other systems," citing the example that "NTR was first developed in mice, but with a prodrug that proved difficult to use, and it was not widely pursued." These two statements are, however, contradictory: if the prodrug proved difficult to use, this constitutes precisely the kind of practical barrier the authors claim does not exist. The authors should clarify and reconcile this inconsistency, and provide a more thorough discussion of why the NTR/Mtz system has seen limited adoption in classical model organisms, such as mice and Drosophila.

---

## [Referee Report · Reviewer #2 (Public review)]

Summary:

Kim and Parsons reviewed the nitroreductase (NTR)/prodrug system: when engineered cells expressing the enzyme NTR are treated with prodrug (e.g. metronidazole), NTR converts the prodrug into cytotoxic compound which kill these cells. The review covers how the system has been developed, spatiotemporal control of targeted cell ablation, and its broad utility to study regenerative mechanisms, model human diseases, and screen chemicals to discover pro-regenerative and protective compounds. They further discussed the newer version of NTR, more potent prodrug, and experimental design, which not only expand the possible utility of the NTR/prodrug system, but allow the research community to develop a precise, reproducible and versatile platform.

Strengths:

The review summarized landmark work application of the NTR/prodrug system, and recent studies in model organisms, with focus on the model organism zebrafish. The review provides a good gateway to understanding the system and considering regenerative studies.

Weaknesses:

None.

Comments on revisions:

The authors have addressed the previous points, and the manuscript has been greatly improved.

---

## [Referee Report · Reviewer #3 (Public review)]

Summary:

This manuscript by Kim and Parsons presents an overview of the nitroreductase/metronidazole (NTR/MTZ) cell ablation system.

Strengths:

This manuscript nicely places the NTR/MTZ system in context of other cell ablation methods, with a discussion of their respective advantages and disadvantages. This review is particularly useful for highlighting the many ways the NTR/MTZ system has been applied to study regeneration of multiple cell types and to model different degenerative human diseases. The review concludes with a discussion on recent improvements made to the system and practical considerations and "best practices" for NTR-based experiments. This review could be a helpful resource, especially for researchers new to regeneration or cell ablation studies.

Comments on revised version:

I thank the reviewers for revising the manuscript to expand their discussion of using the prodrug/NTR system in other model organisms while also revising the abstract to make it clear this review will be zebrafish focused. With these revisions, this review provides an informative overview of how the prodrug/NTR system has not only been an important tool for regeneration studies and but also for elevating the zebrafish as a regeneration model. That said, including other model organisms could have been a nice addition to the last section on experimental considerations, especially in the context of discussing potential barriers to wider adoption of the NTR system. However, given that the vast majority of studies using the NTR system are in zebrafish, the current scope of this review is understandable.

---

## [Author Response]

**Public Reviews:**

**Reviewer #1 (Public review):**
Summary:Kim and Parsons present a timely overview of the NTR/prodrug system and its applications in regenerative biology research, with particular emphasis on tissue-specific cell ablation. The system has substantially advanced the field by enabling non-invasive, conditional cell elimination, and has proven especially powerful in zebrafish, though applications in other classical model organisms are also noted. The review covers the historical origins of the NTR system, its use in regeneration studies, small molecule screening, and genetic and CRISPR-based screening, as well as future directions, including the development of the highly efficient NTR2 enzyme variant.Strengths:This is a useful and well-structured contribution. The manuscript is a valuable resource for the regeneration biology community.Weaknesses:The impact and scientific value of this paper could be meaningfully enhanced by addressing several points outlined below. The concerns centre on completeness, conceptual precision, and the depth of mechanistic discussion.(1) Title: Species specificity.Given that the review's primary focus is the zebrafish model, it would be appropriate to include the species name in the title. This would improve discoverability and accurately set the scope of the article for prospective readers.

Thank you for this suggestion. In revising the review, we have substantially expanded the content to address the reviewers' comments, including adding more detail on the use of NTR in other species. We agree that the majority of published work, and the research we cover, has been conducted in zebrafish, and we have clarified this in the abstract and introduction. However, our aim in writing the review was also to highlight that there is no intrinsic barrier to adopting this technique more broadly in other systems. Notably, NTR was first developed in mice, but with a prodrug that proved difficult to use, and it was not widely pursued. In mouse models, the development of DTR offered an alternative, though that approach carries risks of kidney toxicity and is incompatible with chronic ablation due to immunogenicity. Given this context, we would prefer to retain a title that does not limit the scope exclusively to zebrafish, so as not to discourage readers working in other model systems who might benefit from considering the NTR system.

(2) Subchapter: Physical injury.The subchapter enumerates different types of physical injury models but would benefit from a more substantive comparative discussion. In particular, the authors are encouraged to address the following:(2.1) Outcome comparison: Surgical and other invasive approaches cause damage to entire tissue structures comprising multiple cell types, whereas tissue-specific genetic ablation eliminates a defined cell population while leaving the surrounding architecture largely intact. This fundamental distinction has direct implications for the interpretation of regenerative outcomes and should be clearly articulated.

We appreciate the reviewer raising these important points, as well as those noted in Section 2.2. We addressed the concerns from Sections 2.1 and 2.2 throughout multiple parts of our review, specifically in the following sections:

• Physical injury – where we highlight the importance of precisely characterizing the nature and extent of tissue damage in order to appropriately interpret subsequent biological responses.

• Chemogenetic cell-specific ablation – where we expand on this theme by discussing the advantages of selectively eliminating discrete cell populations and how this improves mechanistic interpretation of regeneration.

• Development of NTR as a suicide gene – where we examine apoptotic pathways and their relevance to nitroreductase-mediated cell ablation.

• NTR/prodrug systems in regenerative studies – where we compare what is currently known about immune activation and inflammatory responses across different NTR-based ablation paradigms.

(2.2) Inflammatory response: Invasive injuries typically trigger a robust inflammatory response, which itself can be a potent driver of regeneration. By contrast, genetic cell ablation may elicit a qualitatively different inflammatory reaction. A comparative discussion of this distinction would help readers appreciate a critical limitation of genetic ablation systems relative to models of natural, accidental tissue damage.

Please see above response 2.1

(3) Subchapter: Cell-specific toxins.This subchapter would benefit from several targeted expansions:(3.1) Off-target effects: The authors should include evidence that the exemplified drugs have known off-target activities, with a discussion of how these confounded the interpretation of experimental data. At least a few concrete published examples should be cited.

Thank you very much for the comments. We have strengthened the discussion of off-target effects by adding concrete published examples. We now note that MPTP/MPP⁺ can affect noradrenergic and serotonergic systems in addition to dopaminergic neurons, that aminoglycoside antibiotics can damage support cells and afferent neurons at higher concentrations with compound-specific differences in ototoxicity, and that streptozotocin exhibits hepatotoxicity beyond pancreatic β-cells.

(3.2) Completeness of the toxin list: The current list appears illustrative rather than comprehensive. A more complete enumeration would be valuable, particularly for neurotoxins and drugs targeting sensory cells, as these are highly relevant to the zebrafish regeneration field.

We have now consolidated the toxins discussed throughout the review into Table 1, which includes additional entries alongside the previously listed agents. We have explicitly noted that this list is representative rather than exhaustive, as the full range of cell-specific toxins used across species is extensive.

(3.3) Interspecies differences: It would be informative to specify whether drug specificity differs across species, as this is a practical consideration for researchers working in organisms other than zebrafish.

We appreciate the reviewer’s question regarding potential interspecies differences in prodrug performance. Early work using NTR in mammals was conducted in mice, and all five published mouse studies relied exclusively on CB1954. No other NTR-activating prodrugs have been reported in mouse models, so direct comparisons are not available. Likewise, all published Xenopus studies used MTZ and thus do not provide internal comparisons across prodrugs. The Nematostella study employed NFP (citing rationale from a zebrafish study) and the approach yielded effective ablation.

The only non-zebrafish study that directly compared prodrugs is the Drosophila work, which evaluated MTZ, RNZ, and NFP and reported lower activity for MTZ relative to the other compounds. Because it is not clear whether the authors were aware of the batch variability of MTZ or the need for freshly prepared solutions, interpreting this specific comparison is difficult.

To address the reviewer’s comment, we have expanded the section on non-zebrafish organisms to clearly state which prodrug was successfully used in each species. However, given the limited number of studies, the absence of titration experiments, and the lack of standardized conditions across laboratories, we do not feel that the available evidence supports drawing conclusions about interspecies differences in prodrug performance.

Consistent with our original discussion and based on the broader biochemical and empirical data available, we continue to recommend RNZ as the starting point for new experiments.

(4) Subchapter: Optogenetic cell ablation.The authors note that optogenetic cell ablation has not yet been applied in conventional regeneration studies. It would strengthen this section to include a discussion of the underlying reasons for this gap, whether technical or biological, so that readers can appreciate the barriers and potential for future adoption.

We thank the reviewer for this helpful suggestion. As recommended, we have added a concise, explicitly speculative statement discussing potential technical factors that may explain why optogenetic cell ablation has not yet been widely applied in regeneration studies. Specifically, we note that KillerRed-based ablation requires localized light delivery and ROS generation, making it best suited for discrete, optically accessible cells and less practical for targeting large or deep tissues. We also highlight that the dependence on microscopy-based illumination inherently limits throughput. This new text clarifies possible barriers to broader adoption while acknowledging that these points remain speculative.

(5) Terminology: "Suicide gene".The use of the term "suicide gene" to nitroreductase is conceptually imprecise and merits reconsideration. Strictly speaking, a suicide gene is one whose expression alone is sufficient to kill the cell, as in the case of genes encoding direct triggers of apoptosis or the catalytic A subunit of diphtheria toxin (DTA). NTR does not meet this criterion: it requires the exogenous administration of a prodrug (e.g., metronidazole) to produce a cytotoxic metabolite and is therefore only conditionally lethal.It is worth noting that nitroreductases evolved in bacteria and fungi as enzymes involved in chemoprotection and detoxification, converting potentially toxic and mutagenic nitroaromatic compounds into less harmful metabolites (PMID: 18355273). This biological context further underscores that NTR is not inherently a lethal protein. The authors are encouraged to replace or qualify the term "suicide gene" and instead adopt terminology that more accurately reflects the conditional, prodrug-dependent nature of the system.

We appreciate the reviewer’s thoughtful attention to terminology. We agree that, in its most classical and stringent sense, a suicide gene is one whose expression alone is sufficient to induce cell death. We also recognize that NTR does not meet this strict criterion.

At the same time, we note that the term has broadened in contemporary usage, particularly within applied and translational contexts, to encompass prodrug-dependent systems. For example, the National Cancer Institute Thesaurus defines a suicide gene as “a gene which will cause a cell to kill itself, typically through interaction with a prodrug,” and Taber’s Medical Dictionary likewise states that it is “a gene that causes a cell to kill itself, usually by encoding an enzyme that converts a nontoxic prodrug into a toxic metabolite.” Under these widely used definitions, NTR is included within the scope of suicide gene systems.

Nevertheless, we appreciate that terminology in this area is not universally standardized. To ensure clarity for all readers, we have added a brief definition in the revised manuscript explicitly noting the conditional, prodrug-dependent nature of NTR-mediated ablation. We are grateful to the reviewer for prompting this clarification.

(6) NTR/MTZ in regenerative studies: Mechanistic depth.

While the review catalogues several studies employing the NTR/MTZ system, it lacks mechanistic depth regarding the cellular basis of ablation. The following questions should be addressed, where evidence exists in the literature:

(6.1) Temporal dynamics of cell death: What is known about the kinetics of NTR/MTZ induced lethality across different tissue types in larval and adult zebrafish, as well as other organisms? Are there age- and tissue-specific differences in the speed or completeness of ablation?

Thank you for this important question. We have added text noting that the kinetics and completeness of NTR/prodrug-mediated ablation vary across experimental contexts, including with differences in NTR expression, enzyme/prodrug pairing, dose, cell type, and developmental stage. Published studies illustrate that the time course of ablation can differ substantially between models. Because most studies were designed to optimize ablation within individual tissues rather than for direct side-by-side comparison, the literature does not yet support broad quantitative conclusions about age- or tissue-specific differences across systems.

(6.2) Mechanism of cell death: What is the cellular basis of NTR/MTZ-induced cytotoxicity in zebrafish? In particular, do the toxic metabolites preferentially cause mitochondrial damage or nuclear DNA damage, and what downstream death pathways are engaged?

Thank you for the comments. We have added text discussing the mechanism of NTR/MTZ-induced cell death. We now note that NTR-mediated reduction of MTZ generates reactive intermediates that cause DNA damage and oxidative stress, with cell death occurring predominantly through apoptosis. We have also more strongly emphasized that in dopaminergic neurons, mitochondrial damage was identified as the primary cytotoxic mechanism. We acknowledge that the relative contribution of these pathways is likely to vary by cell type and remains an important area for future study.

(6.3) Proliferative versus post-mitotic cells: Are proliferating and non-proliferating cells equally sensitive to the NTR/MTZ system, or does the proliferative status of a cell influence susceptibility? This is a practically important question for researchers designing ablation experiments in tissues with mixed cell populations.

We appreciate the reviewer’s insightful question. We have now added a brief clarification to this section explaining that the NTR/MTZ system has been shown to act in a cell-cycle–independent manner, and both proliferating and post-mitotic cells can be ablated effectively.

(6.4) Ablation of progenitor cells: Are there published examples demonstrating that co-ablation of differentiated functional cells and organ-specific progenitor cells abolishes regenerative capacity? Such examples would be highly informative in illustrating the system's power to dissect the cellular requirements for regeneration.

To our knowledge, the zebrafish lateral line currently provides the clearest example in which NTR-mediated ablation of progenitor populations results in a loss of regenerative capacity. In this system, targeted ablation of support-cell progenitors severely reduces hair-cell regeneration, illustrating how NTR enables direct testing of cellular requirements for tissue repair.

Addressing the points above, particularly the comparative discussion of injury models and inflammatory responses, the clarification of terminology, and the mechanistic discussion of NTR/MTZ-induced cell death would substantially strengthen the review's scientific contribution and utility.
**Reviewer #2 (Public review):**
Summary:Kim and Parsons reviewed the nitroreductase (NTR)/prodrug system: when engineered cells expressing the enzyme NTR are treated with prodrug (e.g. metronidazole), NTR converts the prodrug into a cytotoxic compound that kills these cells. The review covers how the system has been developed, spatiotemporal control of targeted cell ablation, and its broad utility to study regenerative mechanisms, model human diseases, and screen chemicals to discover pro-regenerative and protective compounds. They further discussed the newer version of NTR, a more potent prodrug, and experimental design, which not only expands the possible utility of the NTR/prodrug system, but also allows the research community to develop a precise, reproducible and versatile platform.Strengths:The review summarized landmark work application of the NTR/prodrug system, and recent studies, with focus on the model organism zebrafish. The review provides a good gateway to understanding the system and considering regenerative studies.Weaknesses:No weaknesses were identified by this reviewer.
**Reviewer #3 (Public review):**
Summary:This manuscript by Kim and Parsons presents an overview of the nitroreductase/metronidazole (NTR/MTZ) cell ablation system.Strengths:This manuscript nicely places the NTR/MTZ system in the context of other cell ablation methods, with a discussion of their respective advantages and disadvantages. This review is particularly useful for highlighting the many ways the NTR/MTZ system has been applied to study the regeneration of multiple cell types and to model different degenerative human diseases. The review concludes with a discussion on recent improvements made to the system and practical considerations and "best practices" for NTR-based experiments. This review could be a helpful resource, especially for researchers new to regeneration or cell ablation studies.Weaknesses:Although the NTR/MTZ system has been used in other model organisms, this review is primarily focused on its uses in zebrafish. While this is understandable given the wide adoption of NTR/MTZ in the zebrafish field, discussion of the unique considerations and/or challenges for non-zebrafish systems would be an interesting addition and could broaden the potential audience for this review. Additional minor revisions, as suggested below, could also improve readability.
**Recommendations for the authors:**

**Reviewer #2 (Recommendations for the authors):**
Since the lab mouse is an important mammalian model system, with certain tissues harbouring some regenerative capabilities, including the peripheral nervous system (e.g., sciatic nerve regeneration after crush), and myelin, etc., it would be great if a section could be included to discuss the potential adoption of the NTR/prodrug system in future mouse studies.

We appreciate the reviewer’s suggestion to discuss the potential future use of the NTR/prodrug system in mouse models. In surveying the literature, we identified only five mouse studies employing NTR, all of which used CB1954. These early studies were conducted primarily as proof-of-principle work in the context of gene-directed enzyme prodrug therapy (GDEPT) for cancer, rather than for regenerative or lineage-specific ablation applications. We added this point to the text.

Since those reports, we have not found additional examples of NTR use in mice. We do not know the precise reasons for this limited adoption, but it may reflect the availability of alternative ablation systems that are widely established in mouse research, such as the diphtheria toxin receptor (DTR) system.

We agree that certain mouse tissues exhibit regenerative capacity and that targeted ablation tools can be valuable in such contexts. To address the reviewer’s point, we have added text noting the very limited historical use of NTR/CB1954 in mouse. We have no explanation as to why no one moved onto using NTR/MTZ in the mouse but note in two places in the text that DTR is preferred method to use in mouse ablation experiments (even though DT does cause kidney damage and is incompatible with chronic studies!).

Minor:(1) Line 174-176, the sentence was repeated.(2) Figure 1, for the transgenic line, please be consistent with the line name in italics.
**Reviewer #3 (Recommendations for the authors):**
(1) In the abstract as well as in the main text, the authors note that the NTR/MTZ system has been used in multiple model systems. Yet, most of the review, and especially the practical advice given at the end, is very zebrafish-focused. Although this is understandable given the wide adoption of NTR/MTZ in the zebrafish field, the authors might consider revising the abstract to make it clearer that this review is primarily concerned with the use of the NTR/MTZ system in zebrafish.

Thanks for the suggestion. We have changed last half of first paragraph in abstract

That said, a brief discussion of any unique considerations and/or challenges for non-zebrafish systems would be an interesting addition and could broaden the potential audience for this review.

Agreed and we have expanded in several places in the text to discuss more about the NTR system in non-zebrafish. We especially expanded our discussion about NTR in the mouse.

(2) Line 176: There is a repetition of the sentence, "NTR/MTZ-mediated ablation has also been adapted for other model organisms."

Found and deleted. Thank you!

(3) Line 177: To improve clarity, the authors should include species names to prevent confusion. For example, both *Xenopus laevis* and *Xenopus tropicalis* are commonly used model organisms. Similarly, multiple *Drosophila* species are used by researchers.

Added melanogaster and laevis to text.

(4) Can the authors address whether alternatives to MTZ (RNZ, etc.) have the same issues with batch-to-batch variability? That might be an important consideration for potential users. It would also be useful to include practical guidance for accounting for batch variability, for example, how to determine optimal prodrug concentrations, whether effective concentrations need to be determined for every batch/replicate/experiment, etc.

Added text that discusses that, it is not yet known whether RNZ exhibits batch-to-batch variability similar to MTZ, as this has not been systematically reported. Given the potential for variability, it would be prudent for researchers to titrate each new batch of RNZ or, alternatively, adopt a dosing strategy that exceeds the minimum effective concentration to ensure consistent ablation results.

(5) For the last section ("Experimental design: Practical and technical considerations"), readability would be improved by applying a consistent bullet point format.

Made the changes as requested.

(6) Figure 1: Asterisks are not defined.

The asterisks where to link to two boxes depicting the same transgene without rewriting the name of the transgene. Clearly, this wasn’t clear, so we have added explanation to legend too.

(7) Figure 3: Given that the schematics specify expression of NTR1 and NTR1.1, I assume this figure is adapted or based on previous published report(s). If so, the reference(s) should be noted in the figure legend or on the figure itself (as done for Figure 1). If the schematic is meant to depict only in general terms how binary expression vectors can be used, a more inclusive "NTR" label might be less confusing.

Changed figure legend and figure

(8) Figure 4: To improve readability and accessibility, the authors should consider modifying panels C-N to use a more colorblind-friendly palette (e.g., green/magenta) or to present each channel as separate grayscale images.